# A Specific Mixture of Propolis and Carnosic Acid Triggers a Strong Fungicidal Action against *Cryptococcus neoformans*

**DOI:** 10.3390/antibiotics10111395

**Published:** 2021-11-13

**Authors:** Alejandra Argüelles, Ruth Sánchez-Fresneda, Elisa Martínez-Mármol, José Antonio Lozano, Francisco Solano, Juan Carlos Argüelles

**Affiliations:** 1Vitalgaia, España, S.L., 30005 Murcia, Spain; a.arguellesprieto@um.es (A.A.); ruth.sanchez1@um.es (R.S.-F.); 2Área de Microbiología, Facultad de Biología, Universidad de Murcia, 30071 Murcia, Spain; elisa.m.m@um.es; 3Departamento de Bioquímica y Biología Molecular B e Inmunología, Facultad de Medicina, Universidad de Murcia, 30100 Murcia, Spain; jalozate@um.es

**Keywords:** propolis, carnosic acid, fungicides, natural products, *C. neoformans*

## Abstract

Current antifungal chemotherapy against the prevalent basidiomycete *Cryptococcus neoformans* displays some drawbacks. This pathogenic fungus is refractory to echinocandins, whereas conventional treatment with amphotericin B plus 5-fluorocytosine has a limited efficacy. In this study, we explored the potential cryptococcal activity of some natural agents. After conducting a screening test with a set of propolis from different geographical areas, we selected an extract from China, which displayed a certain cytotoxic activity against *C. neoformans*, due to this extract being cheap and easily available in large amounts. The combination of this kind of propolis with carnosic acid in a 1:4 ratio induced a stronger fungicidal effect, which occurred following a synergistic pattern, without visible alterations in external cell morphology. Furthermore, several carnosic acid–propolis formulations applied onto preformed biofilms decreased the metabolic activity of the sessile cells forming biofilms. These data support the potential application of mixtures containing these two natural extracts in the design of new antifungal strategies in order to combat opportunistic infections caused by prevalent pathogenic fungi.

## 1. Introduction

The antifungal therapy currently available in clinical practice does not fully cover the sharp rise in morbidity and mortality caused in recent decades by invasive fungi. This represents a serious concern for public health that mainly affects the immunocompromised population [1,2]. In spite of the arsenal of approved compounds (namely polyenes, azoles, echinocandins, allylamines, and pyrimidine analogues) [3,4] that can be used for the treatment of fungal infections in humans, there are a growing number of isolated nosocomial outbreaks of fungi habitually considered as non-pathogenic recorded in hospitals and a harmful rise in strains resistant to the usual chemotherapeutics. These and other factors increasingly complicate the standardization of effective treatments against invasive mycosis [5].

This scenario is even more worrying regarding the opportunistic basidiomycete *Cryptococcus neoformans*, the etiological agent of AIDS-associated meningoencephalitis, a disease frequently accompanied by respiratory symptoms (e.g., cough, pneumonia, and breathing difficulties) and associated with high mortality (around 20–50% of affected patients [6]). One main problem in getting efficient remedies comes from the intrinsic resistance of *C. neoformans* to echinocandins, due to its very low β-glucan content in the cell wall [6,7]. Furthermore, *C. neoformans* contains other exclusive features of virulence, such as an external polysaccharide capsule [7,8] and the capacity to synthesize melanin in growth media containing diphenolic compounds, such as catecholamines, which are present in the nervous system of infected patients [9]. Melanin is able to absorb and bind a number of drugs in polyphenolic polymer, including polyenes, diminishing the actual doses and efficacy of those treatments in the fungal cells. In addition, the usual treatment with Amphotericin B (AmB) plus 5-Fluorocytosine should be limited to a short time, since long-lasting exposure to conventional AmB causes side effects, notably nephrotoxicity, and the liposomal formulation of AmB is too expensive [10]. The further addition of fluconazole is able to surpass the blood–brain barrier and usually requires prolonged treatments, but even so, it does not always ensure a complete cure [3,4]. 

The empiric therapeutic applications of many natural extracts and semi-purified natural substances endowed with antimicrobial activities come from a long tradition, and they have recently emerged as efficient and alternative clinical treatments [11] because they usually lack unwanted side effects and overcome the problem of microbial resistance to conventional antibiotics.

Among these natural compounds, propolis deserves great attention and represents an important target in biotechnological research. Propolis (here abbreviated as PP) is a complex resinous product collected by honeybees (*Apis mellifera*), which use it as a protective mechanism against insects and other pathogenic organisms [11]. Although the chemical composition of propolis shows great differences depending on environmental and biotic factors, the main components are flavonoids, phenolic acids, terpenes, aromatic aldehydes and alcohols, and fatty acids [12,13,14]. Despite the problems of standardization, propolis samples from diverse geographical origins have been successfully applied as efficient antimicrobial compounds against a number of microbial pathogens, either bacteria or fungi [15,16]. Remarkably, it has recently been reported that ethanolic extracts of propolis display cryptococcal activity by means of the reduction of chitin–chitosan and melanin production [7]. 

In the search for new, potentially interesting, antimicrobial designs based on propolis, we assayed the combination of PP and carnosic acid (CA), a bioactive terpenoid extracted from rosemary (*Rosmarinus officinalis*) and endowed with both antioxidant and antimicrobial activities [17,18,19]. A new, potent antifungal effect triggered by combinatorial mixtures of CA and PP has recently been demonstrated against clinical isolates of the highly prevalent yeast *Candida albicans* [20]. Here, we studied the possible fungicidal action of these formulations against the dangerous and opportunistic basidiomycete *C. neoformans*, whose capacity to develop in vitro biofilms was also analyzed.

## 2. Results

### 2.1. Study on the Antifungal Effect of Some Propolis and CA on C. neoformans

To begin the exploration of the biotechnological applications and therapeutic potential of these natural agents, we carried out an initial screening to check the antifungal action of different commercial propolis extracts (PP) and a semi-purified carnosic acid extract (CA) against growing cells of *C. neoformans* H99. The three propolis extracts selected were obtained from different geographical origins and habitats. PP1 is a Cuban propolis, which according to the literature might be rich in polyprenylated benzophenones as it is formed with organic material from flowers mainly from *Clusia* spp. flora. PP2 and PP3 are two Chinese propolis from different regions which should be rich in flavones and flavanones since they are formed with material mainly from *Populus* spp. [11,12,13]. On the other hand, CA is a semi-purified preparation obtained from rosemary leaves. As shown in Figure 1, not all the propolis examined displayed an intrinsic antifungal action, measured as the percentage of viable cells in comparison to a control (CFU/mL) after 1 h of exposure to the agents. Whereas PP1 was virtually inactive at the concentrations tested, PP2 only displayed conspicuous antifungal action at 1000 µg/mL, and PP3 showed the strongest fungicidal outcome since an intense antifungal action was observed at 500 µg/mL. In turn, CA, even at lower concentrations (50 μg/mL), also displayed a noticeable fungicidal effect against *C. neoformans,* similar to that previously recorded in *C. albicans* [20].

### 2.2. Analysis of the PP3 Extract and Semi-Purified CA by HPLC

Propolis is a heterogeneous crude material whose chemical composition is extremely complex and depends on many biotic and abiotic conditions. The results presented in Figure 1 suggest that it is not feasible to use a unique standard formula of propolis to induce a universal outcome against a specific pathogen. Indeed, the effect of Chinese propolis PP3 was much higher than the batch of Cuban propolis tested, although at identical concentrations (50 μg/mL) the fungicidal action triggered by PP3 was relatively weak as compared to that of CA. However, this effect was stronger at concentrations over 500 μg/mL (Figure 1).

Taking into account the absence of important side effects of natural agents, it seems necessary to further explore their composition and antifungal properties. Therefore, we analyzed the most powerful batch tested (PP3) by HPLC using DAD-UV detection. Figure 2A displays the profile components of an ethanolic extract of this PP3 extract at 280 nm. This wavelength allows the general detection of aromatic components, both phenols and flavonoids. The insert shows the same chromatographic profile followed at 340 nm, which was chosen because some flavonoids (mostly flavonols and flavones) also show a second absorption peak in the region of 300–370 nm and display a higher absorption in that region in comparison to phenolic esters. The chromatographic profile corresponding to the semi-purified CA is shown in Figure 2B. These chromatograms reveal the complexity of the propolis composition, although the CA extract, whose calculated purity is around 70–72%, is much less complex and mainly contains this component.

### 2.3. Antifungal Activity of CA and PP Mixtures on C. neoformans

As stated above, the antifungal effects of PP3 became noticeable at concentrations over 500 µg/mL. A combined therapy of propolis and some clinical synthetic antifungals was successfully applied against various pathogenic fungi. For instance, mixtures containing propolis plus Amphotericin B or azoles caused a drastic cell killing in *C. albicans* [21]. We recently demonstrated that a specific combination of PP3 and CA possesses a strong fungicidal effect against *C. albicans*, whether in laboratory or in clinical strains [20]. However, limited information is available on equivalent formulations combining propolis together with other natural compounds on different fungal species. Therefore, we examined whether a similar action might be operative against *C. neoformans.* Furthermore, the hypothetical fungicidal effect after exposure to different times was also monitored. 

The time-course kinetics of cell survival recorded in growing H99 cells revealed that the combinatorial addition of CA and PP3 induced a stronger fungicidal action in comparison with the action exerted by the isolated components at the tested doses. The combination ratio was 1:4 (50 μg/mL CA and 200 μg/mL PP3), which was the optimal ratio already established for *C. albicans* [20]. Consistent results were obtained by either direct counting of viable cells in liquid media (CFUs/mL) (Figure 3A) or by macroscopic colonial formation on solid plates (Figure 3B). It should be noted that although this decay in viability was maintained over at least 5 h, *C. neoformans* cells were able to resume active growth after longer incubations with PP3 (24 h) (Figure 3), indicating that the effect of propolis per se might be fungistatic rather than fungicidal [8].

On the other hand, we also calculated the MIC_50_ for CA and PP3 upon exposure to the *C. neoformans* H99 strain, following the protocol established by the European Committee on Antimicrobial Susceptibility Testing (EUCAST) [22]. The corresponding values were 62.50 μg/mL for CA and 375 μg/mL for PP3. Whereas these concentrations are within the range previously reported for CA extracts in other *C. neoformans* genetic backgrounds, the MIC_50_ of PP3 was higher than the values determined for some American propolis, such as Argentinean [23], but clearly lower with respect to other Asiatic (Korean) propolis [24]. The fractional inhibitory concentrations index (FICI) calculated from the corresponding MIC_50_ was 0.49. This result, together with the isobolograms also performed (Appendix A), provided consistent evidence that the combined mixture of CA and PP3 shows strong synergy [10,20].

The inspection by optical microscopy upon exposure to the CA and propolis extracts did not reveal significant changes in cellular morphology (Figure 4). Although both a weak swelling and an increase in inner granularity could be observed in some formalin-fixed cells by the Nomarski interferential contrast (Figure 4; arrows), the results were not absolutely conclusive. In addition, the formation of the external polysaccharide capsule was not apparent in this analysis.

### 2.4. The Capacity to Form Biofilms by C. neoformans Is Impaired by the Addition of CA and PP3

Like other microbial pathogens, invasive infections caused by *C. neoformans* can progress through the formation of highly structured biofilms, a main factor of virulence, and are especially dangerous among hospitalized immunocompromised patients [25]. The sessile cells responsible for the attached microbial communities frequently become refractory to treatments of conventional antibiotics, including antifungal compounds. In turn, the counterpart planktonic free-living cells become more sensitive. A derived clinical problem comes from the fact that the alternative therapeutic options available to successfully combat the drug-resistant biofilms are limited due to several well-known factors of this fungal arrangement [26,27,28,29].

In accordance with Figure 5, in comparison to an untreated control, the addition of increasing combined concentrations of CA and PP induced a significant reduction in the level of biofilms preformed in vitro during a 24 h period. The percentage of viability in these sessile cells was determined by means of the XTT reduction assay described in the Methods section. Following a previous approach [20,30], several doses of CA were fixed with the concurrent inclusion of growing doses of PP. Although the impairment in the degree of biofilm production during an additional 24 h occurred in a concentration-dependent manner of the chosen mixtures, a greater toxic effect of CA in comparison to PP3 extracts can be seen, reaching the lower metabolic capacity of H99 sessile cells at 500/1000 μg/mL of CA (Figure 5). These results reinforce the importance of synergistic CA–PP3 formulations as a promising target in the design of future antifungal therapies. They also suggest the need of new clinical trials introducing these natural extracts on patients affected by cryptococcosis. 

## 3. Discussion

Some substances obtained from distinct natural sources have been successfully introduced in antimicrobial therapies as a safer alternative to synthetic drugs, or at least as a complementary component of chronic treatments. Among these natural agents with beneficial antimicrobial effects, propolis is a traditional remedy in folk medicine throughout the world and there is a great number of reports supporting the idea that propolis possesses antimicrobial, antioxidant, and immunostimulating activities [11,12,15,16]. In particular, a moderate fungicidal activity was reported against *C. albicans* and *C. neoformans* [31,32]. However, the majority of these observations were performed in vitro by means of a variety of assays using distinct types of propolis. Therefore, the results are far from conclusive. A main drawback was the lack of a constant and homogeneous standard propolis composition, which can be influenced by the climate, environmental plant habitat, or even the genetic variability of the *A. mellifera* “queen bee” forming the product. As a matter of fact, the activity of the three propolis extracts tested herein against *C. neoformans* displayed large differences (Figure 1). 

Propolis usually contains a variety of different chemical components. Currently, more than 300 compounds have been identified so far using the HPLC–DAD–MS method for the quantification of raw ethanolic extracts [33,34]. Most of them are phenolic acids and esters and flavonoids, but other families such as essential oils, terpenes, stilbenes, and beta-steroids have also been detected. The relative proportions of constituents seem to essentially depend on the geographic origin [35]. European and Chinese propolis are usually characterized by a high content of flavonoids, while some American propolis contain a lower amount of this family and more abundance of polyprenylated benzophenones and esters of p-coumaric, caffeic and quinic acids [11,12]. Some studies propose that the high content of flavonoids could be used as a “quality marker”, although cinnamoyl and caffeoyl esters can have various positive effects on human health [32,33,34]. We found that our batch of Cuban propolis was almost ineffective (Figure 1), owing to its low content in flavonoids in comparison to Chinese propolis. However, some recent reports regarding Brazilian and Paraná propolis suggest that American propolis certainly also have antifungal properties [36,37].

The available literature [33,35] indicates that HPLC fractionation of ethanolic extracts obtained from diverse propolis give place to a similar elution profile of phenols and flavonoids in spite of the slight variations due to the different HPLC column, mobile phase, or hydrophobic gradient used. The elution order occurs according to the hydrophobicity of the components, with the more hydrophobic compounds being eluted at longer retention times (Rt) in the interval of 30–55 min for chromatographies lasted around 1 h like this one. According to these results and our own observations (Figure 2A), the first active compounds eluted are flavanols and some caffeoyl acid esters, followed by flavanones, such as pinocembrin; flavones, such as apigenin or chrysin; and flavonols, such as quercetin or galangin. Most of those flavonoids were identified at the peaks in Figure 2A,B by comparison of the total Vis–UV spectra with authentic standards (data not shown). Consequently, a complete MS identification is underway for the definitive characterization of the most active components of raw PP3 propolis extracts responsible for antifungal activity in combination with CA.

Alternative fungicidal treatments based on formulations containing a combination of potentially useful compounds have received great attention. Thus, mixtures of propolis plus azoles (fluconazole or voriconazole) or propolis plus AmB displayed strong toxic action against several *Candida* spp. [21]. In this study, a novel combination consisting of CA and a selected PP3, whose previous efficacy against *C. albicans* has been demonstrated [20], was tested against *C. neoformans*. According to our results, the addition of mixtures containing CA and PP3 in a ratio 4:1 caused a high degree of cell killing on actively growing cells of *C. neoformans* compared to the effect recorded by individual compounds (Figure 3), without any appreciable damage on the cell wall surface (Figure 4). Under these experimental conditions, the formation of an external capsule was not evident. Notably, the application of this CA–PP3 mixture conveys a synergistic effect (measured by the FICI) [10]. Likewise, ethanolic extracts of Polish propolis with distinct azoles showed a similar synergism against various species of *Candida* [21]. However, the capacity of this formula to inhibit the formation of active biofilms could also be recorded at high concentrations of CA (Figure 5), which is a serious limitation for its further clinical application.

There is a wide scope for screening multiple combinations of different types of potentially antifungal substances. For instance, a recent study proved the existence of marked synergism between AmB and off-patent drugs against *C. neoformans* and *Candida* spp. [10]. In our view, however, this strategy has some risks since it is generally assumed that the application of natural compounds in medical practice lacks the undesirable side effects induced by conventional pharmaceutical compounds. We think this idea must be handled with caution. The hypothetical mutagenic effects of propolis should not be discarded, although its antimicrobial activity is apparently not associated with direct DNA damage [33]. Regarding the mechanisms accounting for the antifungal action of propolis and CA, the whole picture is not yet complete. It is known that flavonoids and CA are able to exert a combined action on ROS formation and the inhibition of the ergosterol synthesis, leading to subsequent damage in the membrane of *C. neoformans*. However, the effect of other components still remains elusive. Undoubtedly, this area deserves further and more intensive research.

## 4. Materials and Methods

### 4.1. Yeast Strains and Culture Conditions

The *C. neoformans* H99 standard strain was kindly provided by Dr. O. Zaragoza (Instituto de Salud Carlos III, Madrid, Spain) and used throughout this study. Yeast cell cultures were grown at 37 °C by being shaken in a YPD medium consisting of 2% peptone, 1% yeast extract, and 2% glucose. The strain was refreshed from a −80 °C glycerinated stock and maintained at 4 °C by periodic subculturing in solid YPD or Sabouraud dextrose medium. Solid media contained 2% agar.

### 4.2. Natural Extracts

Fresh leaves of rosemary (*Rosmarinus officinalis*) and common sage (*Salvia officinalis*) were provided by Naturitas, Spain and used to obtain semi-purified extracts enriched in CA [19]. As reported elsewhere [20], they were further enriched to around 72%. Purity was verified by HPLC connected to a Diodo-Array Agilent UV–Vis detector using the conditions described below. This extract was dissolved in 98% ethanol (Panreac, Applichem) until the required concentration for the antifungal assays was reached [20].

Three propolis extracts (PP) were obtained from local suppliers to be initially tested. These extracts were collected from different continental regions in Cuba and China. The preparation of the extracts as well as the analytical details were described by Argüelles et al. [20]. The raw PP stuffs were ground to a fine powder and dissolved by gentle shaking in prewarmed ethanol (98%, at 55 °C) at the required concentrations for antifungal assays. A control with 98% ethanol was run without any significant antimicrobial effect. Other samples were dissolved in dimethyl sulfoxide (DMSO, Sigma-Aldrich, St. Louis, MO, USA) for chromatographic analysis (see below). The estimated total polyphenols content was in the 70–90% range (dry weight). 

### 4.3. HPLC

Conditions for analytical HPLC were as described elsewhere [38] with slight modifications. Samples of semi-purified CA or raw PP (5 mg/mL) were dissolved in DMSO and filtered through a nylon membrane with a 0.45 mm pore size. Amounts of 20 µL of solution were injected into a LiChrospher 100-C18 reverse-phase column (250 × 4.0 mm inner diameter) thermostatized at 30 °C. The mobile phase consisted of a gradient from acetonitrile/2.5% acetic acid aqueous solution starting with 5%/95% and finishing with 95%/5% acetonitrile/acidic water. The flux was 1 mL/min. Detection was carried out by a DAD-UV detector (Agilent, Santa Clara, CA, USA). Usually, chromatographic profiles were monitored at 280 and 340 nm for the differentiated identification of phenols and flavonoids, but complete spectra of some selected major peaks could be analyzed for the identification of selected components.

### 4.4. Determination of Cell Viability

Identical aliquots of *C. neoformans* H99 grown at 37 °C in YPD until the exponential phase (OD_600_ nm = 0.8–1.0) were treated with several concentrations of CA and PP. Cell viability was determined in samples diluted appropriately by plating in triplicate on solid YPD after incubation for 1–2 days at 37 °C [26]. Survival percentages were normalized to control samples (100% viability). Colony growth in solid medium was tested by spotting 5 μL from the respective tenfold dilutions onto YPD agar. Then, the plates were incubated at 30 °C and scored after 48–72 h. Amphotericin B (AppliChem GMBH, Darmstadt, Germany) was included as a positive control of antifungal activity [26,27].

### 4.5. Morphological Analysis

After exposure to the different antifungals, cell morphology was recorded with a Leica DMRB microscope using the Nomarski interference contrast technique. The microscope was equipped with a Leica DC500 camera connected to a PC containing the Leica Application Suite V 2.5.0 R1 software (Wetzler, Germany).

### 4.6. Biofilm Formation

The analysis of in vitro biofilm formation on the surface of polystyrene 96-well microtiter plates utilized previously described methods [28]. Biofilm formation was prepared from *C. neoformans* suspension (1.0 × 106 blastoconidia/mL) in RPMI 1640 at 37 °C for 24 h. Then, CA and PP were added immediately either as separate extracts or as mixtures, and the biofilms were further incubated for an additional 24 h. The quantification of biofilms was carried out as reported elsewhere [29] using the 2,3-bis-(2-metoxy-4-nitro-5-sulfophenyl)-2H-tetrazolium-5-carboxalinide reduction assay (XTT) (Sigma Chemicals, St. Louis, MO, USA). Data were expressed as the percentage of metabolic activity in the treated biofilm samples compared to 100% (untreated controls).

## 5. Conclusions

The results presented in this study demonstrate a marked antifungal effect of carnosic acid and a specific Chinese propolis (so-termed PP3) against the invasive basidiomycete *C. neoformans*. They also suggest that specific combinations of tested natural bioactive substances with antifungal antibiotics (mainly polyenes and azoles) can improve the available clinical therapy to combat highly prevalent pathogenic fungi. New research efforts are necessary in order to characterize the active molecules responsible for the fungicidal effect triggered by propolis as well as the mechanisms involved in the synergistic action between CA and PP3.

## 6. Patents

A patent resulted from the work reported here. Title: Synergistic composition comprising propolis and carnosic acid for use in treating and preventing infections caused by species of the *Cryptococcus neoformans* fungus. Authors. J.A. Lozano, J.C. Argüelles, A. Argüelles, R. Sánchez-Fresneda, J.P. Guirao-Abad and B. Alburquerque. Reference: Europe (EP 3278798). Date of concession: 10/301/2019.

## Figures and Tables

**Figure 1 antibiotics-10-01395-f001:**
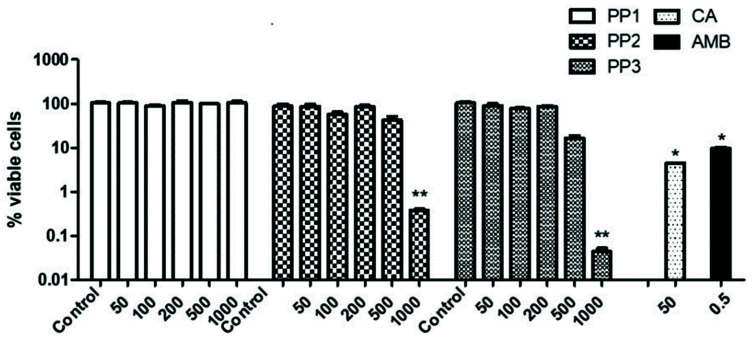
Cellular survival of *C. neoformans* H99 strain recorded upon exposure to several extracts of propolis or CA. Cells were grown on YPD until exponential phase (O.D. = 0.8–1.0). Samples of identical cell density (10^7^ cells/mL) were exposed for 1 h at 37 °C to crude extracts of the indicated propolis in a range of 50–1000 µg/mL as well as to 50 µg/mL of carnosic acid. An untreated control was always run in parallel. Amphotericin B (AmB at 0.5 µg/mL) was introduced as positive control of fungicidal activity. Statistically significant differences * : *p* < 0.05; ** : *p* < 0.01 with respect to an untreated control were tested using the Mann–Whitney U test.

**Figure 2 antibiotics-10-01395-f002:**
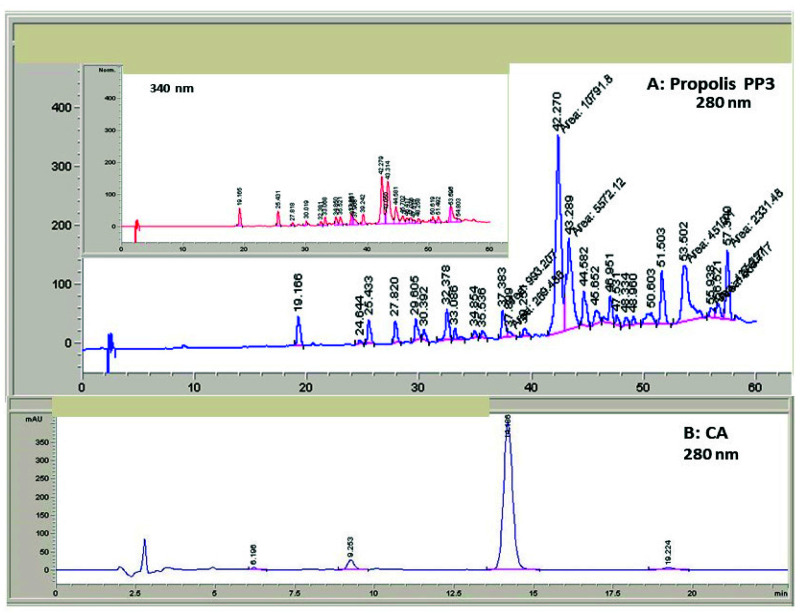
HPLC chromatographic profiles of propolis PP3 and CA monitored by absorbance at 280 nm (**A**,**B**) and 340 nm (**A** insert). For details on the specific protocols followed for sample preparation and HPLC conditions, see Methods.

**Figure 3 antibiotics-10-01395-f003:**
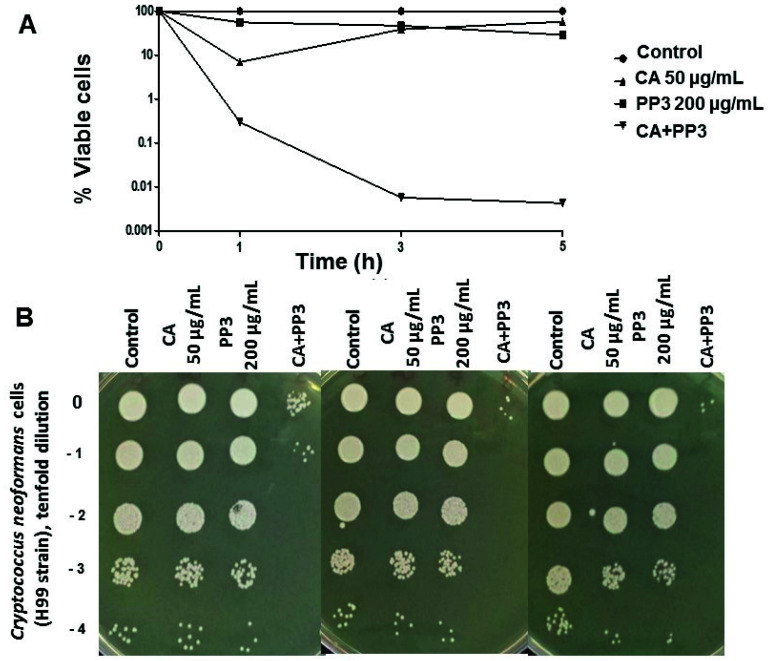
Time-course evolution of cell viability (%) after the addition of 50 μg/mL CA and 200 μg/mL PP3, either separately or in a mixture, to cultures of *C. neoformans* H99 growing on YPD. Exponential cells were aliquoted and treated for 1, 3, and 5 h at 37 °C with the indicated concentrations of CA and PP3. After incubation, identical samples were harvested, washed, and the percentage of surviving cells determined in liquid medium by CFU counting (**A**). The macroscopic colonial growth in the tested conditions was also recorded by spotting tenfold serial dilutions onto YPD plates and was further incubated at 37 °C for 24 h (**B**). The data shown are representative of three independent experiments. Statistically significant differences with respect to an untreated control were tested using the Mann–Whitney U test.

**Figure 4 antibiotics-10-01395-f004:**
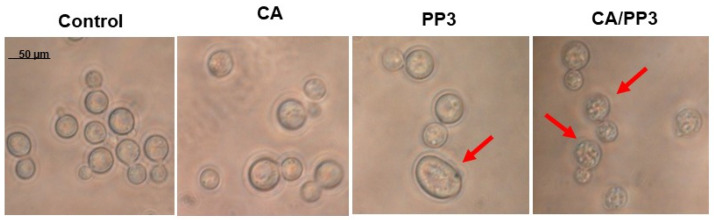
Morphological analysis of *C. neoformans* H99 yeast-like cells after treatment with separate additions of CA (50 μg/mL), PP3 (200 μg/mL) or a combination of both extracts. YPD-grown exponential cells (OD_600_ = 0.9) were exposed for 5 h at 37 °C to the indicated concentrations of CA and PP. An untreated sample was maintained as a control. Two similar representative images from each treatment were taken by means of Nomarski interferential contrast. The meaning of red arrows is explained in the text.

**Figure 5 antibiotics-10-01395-f005:**
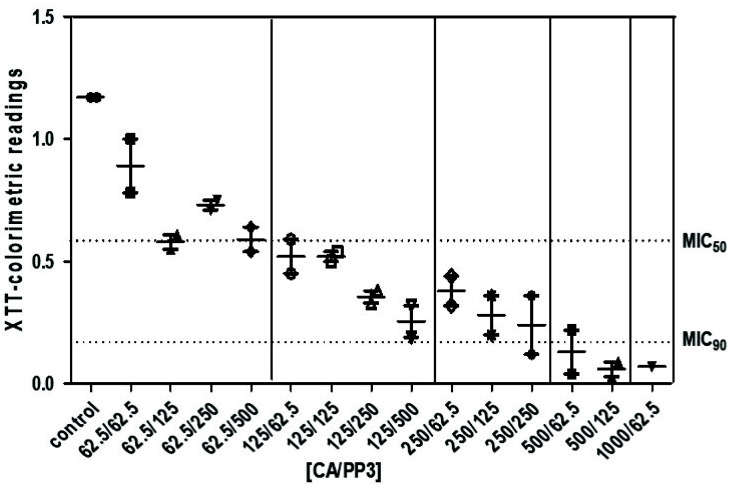
Inhibitory effect of different mixtures of CA and PP on the viability of preformed biofilms recorded in *C. neoformans* H99 strain. Planktonic cells were allowed to adhere on the surface of polystyrene 96-well microtiter plates at 27 °C for 24 h. After addition of the extracts during further 24 h, the metabolic activity of these preformed biofilms was quantified by the XTT reduction assay (see Methods). The results are expressed as the mean ± standard deviation of two experiments with six replications for each group. The concentrations of CA and PP are expressed as µg/mL.

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
