# Peer review of "A Specific Mixture of Propolis and Carnosic Acid Triggers a Strong Fungicidal Action against *Cryptococcus neoformans"

_antibiotics, 2021, doi:10.3390/antibiotics10111395_

Round 1

Reviewer 1 Report

Correctly and interestingly written article describing the synergistic effect of propolis and carnosic acid against Cryptococcus neoformans. For me, the article is a continuation of the research of the same team from ref. [20]. Sufficient introduction, well-conducted discussion and presentation of results. It is a pity that the authors did not use the HPLC-MS method to analyze the composition of individual propolis. Missing section "Conclusions".

Remarks:
1) There should be "mL" throughout the article, not "ml"
2) Throughout the article, the Latin names of plants or strains should be written in italic.
3) Please add the producers, cities and countries of the chemicals and equipment used.
4) There is no "Conclusions" section.

Minor revionsions.

Author Response

Reviewer #1

We truly appreciate the suggestions made by this reviewer, which are essentially supportive. However, he/she regrets the lack of a more detailed analysis of the individual propolis extract used here. We concur with this view. In fact, a determination of propolis composition by HPLC-MS has already accomplished, but the results are considered an “industrial property” of Vitalgaia España, S.L. and they cannot be published including all details.

Remarks:

1) There should be "mL" throughout the article, not "ml"… 2) Throughout the article, the Latin names of plants or strains should be written in italic.

Indeed, these suggestions contribute to improve the scientific style of our manuscript. Unfortunately, the nomenclature was correct in the word format, but sometimes suffered a wrong reconfiguration during the conversion for submission. The volumes (mL instead of ml) and Latin names of biological species are now properly written and labelled in red at the revised version of the manuscript.

3) Please add the producers, cities and countries of the chemicals and equipment used.
This has been done.

4) There is no "Conclusions" section.

A new section (5) devoted to the main Conclusions collated from the experimental research has been included.

Reviewer 2 Report

Lines 51-52: “since AmB has  some side effects including certain toxicity in long lasting treatments,” is too vague. The authors should explicitely describe the key safety issues that limit the use of AmB.

Lines 53-55: This statement needs to be accompanied by at least an appropriate reference.

Lines 87-91: These statements about the chemical composition by source should be supported by appropriate references.

Line 97: an MIC of 500 ug/ml seem too high to be considered actually active, the authors should justify why at such a level the extract was considered active.  

Line 164: „This result together with the isobolograms also performed (results not shown)” – the isobolograms should at least be made available as Supplementary electronic materials.

Lines 215-216: the concentrations proposed there (500/1000 μg/ml of CA) are too high to be of clinical relevance in our view. This should also be acknowledged as an important limitation in the discussion section.

Lines 241-243: this sentence should be accompanied by at least one supporting reference. The same hold strue for the sentence from lines 244-245.

Lines 257: „flavonololols” should be corrected to „flavonols”

Line 301: there is no such thing as an “extract of CA”, maybe an extract enriched in/containing CA.

Line 302: “around 72%” is pretty vague, using two decimals and stating the assay method is preferrable.

Line 313: are the percentages (70-90%) reported on a dry basis?

Author Response

Reply to Reviewer # 2

The evaluator thinks that there is room for a further revision and better clarification of the manuscript, and points out the necessity of introducing some changes in the English language. We believe the reviewer is right and agree with those points. Thus, we have carried out some modifications. These are point by point our answers to the queries raised.

Comments and Suggestions for Authors

Lines 51-52: “since AmB has some side effects including certain toxicity in long lasting treatments,” is too vague...

The reviewer is right. The corresponding brief phrase is now rewritten to read: "since long lasting exposure to conventional AmB cause side-effects, notably nephrotoxicity”.

Lines 53-55: This statement needs to be accompanied by at least an appropriate reference.

  1. This statement is supported by the set up references 3 and 4.

Lines 87-91: These statements about the chemical composition by source should be supported by appropriate references.

This is a good remark. A precise composition of each propolis extracts acquired from commercial sources is not available from suppliers. We have estimated an approximate composition based on the geographic regions and the plants described in the literature. In accordance with the reviewer, three new references are cited: 11-13. Please note these references were in the original reference list.

Line 97: an MIC of 500 ug/ml seem too high to be considered actually active, the authors should justify why at such a level the extract was considered active.

There seems to be some confusion here. We do not report a MIC of 500 μg/ml, but we stated that an intense antifungal effect was observed at that concentration for propolis PP3. Although it would be feasible to calculate a MIC for a complex extract with a large number of components, this parameter is more accurate for single, purified compounds. We believe this phrase has to be maintained in its current form.

Line 164: “This result together with the isobolograms also performed (results not shown)” – the isobolograms should at least be made available as Supplementary electronic materials.

The claim of the reviewer will be satisfied. The presence of an isobologram provides additional support on the proposed synergistic effect of carnosic acid and propolis on C. neoformans. Therefore, a Supplementary Figure 1 has been included.

Lines 215-216: the concentrations proposed there (500/1000 μg/ml of CA) are too high to be of clinical relevance in our view. This should also be acknowledged as an important limitation in the discussion section.

We agree with the reviewer concerning the severe limitations in clinical therapy of these high concentrations. However, they were exclusively used in a correlative assay of both natural extracts during biofilm formation, which confirm the toxicity of carnosic acid.

In accordance, an explanatory sentence regarding this limitation was introduced in the Discussion: “However, the capacity of this formula to inhibit the formation of active biofilms could also be recorded at high concentrations of CA (Figure 5)…”.

Lines 241-243: this sentence should be accompanied by at least one supporting reference. The same hold true for the sentence from lines 244-245.

  1. The appropriate supporting references are mentioned as indicated.

Lines 257: “flavonololols” should be corrected to “flavonols”

  1. This mistake has been corrected.

Line 301: there is no such thing as an “extract of CA”, maybe an extract enriched in/containing CA.

This expression is currently modified as “an extract enriched in CA”

Line 302: “around 72%” is pretty vague, using two decimals and stating the assay method is preferable.

Unlike synthetic products, the great variability recorded in natural substances makes difficult the introduction of decimals. The quantitation was determined using an integration of the HPLC peaks. We certainly believe the term “around 72%” is sufficiently informative, but we cannot provide a precision including decimal figures.

Line 313: are the percentages (70-90%) reported on a dry basis?

Yes, they are. We clarify this point in the manuscript “The percentage 70-90 (dry weight)”

Round 2

Reviewer 2 Report

In my view, the manuscript has now been improved and it could be published.